# Clinicopathological-Associated Regulatory Network of Deregulated circRNAs in Hepatocellular Carcinoma

**DOI:** 10.3390/cancers13112772

**Published:** 2021-06-02

**Authors:** Jian Han, Thomas Thurnherr, Alexander Y. F. Chung, Brian K. P. Goh, Pierce K. H. Chow, Chung Yip Chan, Peng Chung Cheow, Ser Yee Lee, Tony K. H. Lim, Samuel S. Chong, London L. P. J. Ooi, Caroline G. Lee

**Affiliations:** 1Department of Biochemistry, Yong Loo Lin School of Medicine, National University of Singapore, Singapore 119077, Singapore; a0123673@u.nus.edu; 2NUS Graduate School for Integrative Sciences and Engineering, National University of Singapore, Singapore 119077, Singapore; thomas.thurnherr@u.nus.edu; 3Department of Hepato-Pancreato-Biliary & Transplant Surgery, Singapore General Hospital, Singapore 169608, Singapore; alexander.chung.y.f@singhealth.com.sg (A.Y.F.C.); brian.goh@singhealth.com.sg (B.K.P.G.); pierce.chow.k.h@singhealth.com.sg (P.K.H.C.); chan.chung.yip@singhealth.com.sg (C.Y.C.); cheow.peng.chung@singhealth.com.sg (P.C.C.); seryee.lee@duke-nus.edu.sg (S.Y.L.); london.lucien.ooi.p.j@singhealth.com.sg (L.L.P.J.O.); 4Cancer and Stem Cell Biology Program, Duke-NUS Graduate Medical School Singapore, Singapore 169547, Singapore; 5Department of Surgical Oncology, National Cancer Centre Singapore, Singapore 169610, Singapore; 6Department of Pathology, Singapore General Hospital, Singapore 169608, Singapore; lim.kiat.hon@singhealth.com.sg; 7Department of Pediatrics, Yong Loo Lin School of Medicine, National University of Singapore, Singapore 119077, Singapore; samuel_chong@nuhs.edu.sg; 8Division of Cellular & Molecular Research, Humphrey Oei Institute of Cancer Research, National Cancer Centre Singapore, Level 6, Lab 5, 11 Hospital Drive, Singapore 169610, Singapore

**Keywords:** hepatocellular carcinoma, circular RNA, clinical characteristics, carcinogenesis, competing endogenous RNAs

## Abstract

**Simple Summary:**

Here, we present a novel strategy to identify key signatures of clinically-relevant co-expressed circRNA-mRNA networks in pertinent cancer-pathways that modulate the prognosis of HCC patients, by integrating clinicopathological features, circRNA and mRNA expression profiles. Five master circRNAs were identified and experimentally demonstrated to upregulate proliferate and promote transformation. Through further integration with miRNA-expression profiles, clinically-relevant competing-endogenous-RNA (ceRNA) networks of circRNA-miRNA-mRNAs were constructed. The most up-regulated nodal-circRNA, circGPC3 was experimentally demonstrated to up-regulate cell-cycle, migration and invasion. circGPC3 was found to act as a sponge of miR-378a-3p to regulate ASPM expression and modulate cell transformation. These 5 nodal circRNAs has potential to be good prognostic biomarkers with good prognostic performance. circGPC3 has great potential to be a promising non-invasive prognostic biomarker for early HCC. We have thus demonstrated the robustness of bioinformatically-predicted master circRNAs in clinically-relevant, circRNA-mRNA networks, underscoring the important roles that these identified deregulated key/master circRNAs play in HCC.

**Abstract:**

Hepatocellular carcinoma (HCC) is one of the most common and lethal cancers worldwide. Here, we present a novel strategy to identify key circRNA signatures of clinically relevant co-expressed circRNA-mRNA networks in pertinent cancer-pathways that modulate prognosis of HCC patients, by integrating clinic-pathological features, circRNA and mRNA expression profiles. Through further integration with miRNA expression profiles, clinically relevant competing-endogenous-RNA (ceRNA) networks of circRNA-miRNA-mRNAs were constructed. At least five clinically relevant nodal-circRNAs, co-expressed with numerous genes, were identified from the circRNA-mRNA networks. These nodal circRNAs upregulated proliferation (except circRaly) and transformation in cells. The most upregulated nodal-circRNA, circGPC3, associated with higher-grade tumors and co-expressed with 33 genes, competes with 11 mRNAs for two shared miRNAs. circGPC3 was experimentally demonstrated to upregulate cell-cycle and migration/invasion in both transformed and non-transformed liver cell-lines. circGPC3 was further shown to act as a sponge of miR-378a-3p to regulate APSM (Abnormal spindle-like microcephaly associated) expression and modulate cell transformation. This study identifies 5 key nodal master circRNAs in a clinically relevant circRNA-centric network that are significantly associated with poorer prognosis of HCC patients and promotes tumorigenesis in cell-lines. The identification and characterization of these key circRNAs in clinically relevant circRNA-mRNA and ceRNA networks may facilitate the design of novel strategies targeting these important regulators for better HCC prognosis.

## 1. Introduction

Hepatocellular carcinoma (HCC) is the commonest liver cancer, representing 70–85% of all cases [1] and is the fifth commonest cancer type with a dismal prognosis of ~18% 5-year overall survival [2], mainly due to late diagnosis, high rates of postoperative recurrence, metastasis and paucity of early diagnosis biomarkers. Chronic hepatitis-B/C viruses (HBV/HCV) are major risk factors for HCC [3].

Recent high-throughput RNA sequencing reveals that more than 70% of the genome is transcribed to ncRNAs [4,5]. circRNAs are a class of ncRNAs produced by a back-splice event from pre-mRNA [6] and can be classified into five groups according to the relationship with coding RNA in the transcript: “exonic”, “intronic”, “intergenic”, “sense” and “antisense” [7]. While circRNAs can function in different ways at the molecular level, current evidence suggests that circRNAs, particularly exonic circRNAs, mainly act as sponges for miRNA harboring miRNA response elements (MREs) to affect the activity of miRNA-target mRNA interaction [8]. For instance, circMTO1 suppresses HCC progression by sponging oncogenic miR-9 to enhance p21 expression, attenuating HCC cell proliferation and invasion [9]. circRNAs can also act as protein sponges [10] modulating protein function [11], or act as scaffolds for protein [12] recruiting proteins to specific locations [13], or undergo cap-independent translation [14]. As circRNAs are conserved, stable, abundant and diverse [15], they have potential to serve as diagnostic/prognostic biomarkers.

Current studies mainly focus on single circRNA inferred from host genes or selected from deregulation profiles before their tumorigenic roles and/or clinical association were investigated [9,16]. A few studies identified/catalogued lists of circRNAs that were deregulated in only a few HCC patients, identifying 26–87 differentially expressed circRNAs in HCC patients [17,18]. A database of circRNAs was also collated from sequencing data of 5 HCC patients [19]. In addition, circRNA-based competing RNA (ceRNA) networks from limited publicly available datasets comprising between 3 and 7 HCC patients were also constructed [20,21].

As several circRNAs may work together to modulate tumorigenesis, metastasis or even patients’ prognosis, it is thus pertinent to identify and characterize networks of circRNA comprehensively as these may function synergistically. Here, we employed a novel network-based strategy to identify clinically relevant, differentially expressed key/master cirRNA regulators through global integration of deregulated cirRNAs, miRNAs and mRNAs with clinical characteristics and pertinent cancer pathways.

## 2. Materials and Methods

### 2.1. HCC Patients and Clinical Samples

Tumorous tissue, adjacent non-tumorous liver tissue and blood of HCC patients from Singapore General Hospital were obtained after informed consent and prior approval from the SingHealth Institutional Review Board (SingHealth CIRB Ref: 2018/3155) and kept in liquid nitrogen. All methods were carried out in accordance with relevant guidelines and regulations, and the tissues were anonymized to the bench researchers. The demographics and clinical characteristics of these HCC patients are summarized in Appendix A. The circRNA, miRNA and mRNA expression profiles of 49 HCC patients’ tumorous tissue and adjacent non-tumorous liver tissue were obtained via microarray or sequencing. The expression profiles for circRNAs, miRNAs and mRNAs in this study were deposited in Gene Expression Omnibus with series entries GSE155949, GSE156087 and GSE138178, respectively.

### 2.2. Workflow for Analyses of Clinically Relevant circRNA Networks in HCC

The strategy to identify clinically relevant circRNA-mRNA and circRNA-miRNA-mRNA competing RNA (ceRNA) networks from circRNA, miRNA and mRNA expression profiles of 49 HCC patients is presented in Figure 1A.

Clinically relevant deregulated circRNA-mRNA pairs are identified as follows: (1) Significantly deregulated circRNA (|FC| > 1.5, FDR < 0.05) are associated with various clinical characteristics (Appendix A) using either a Student’s t-test for binary characteristics or a log-ranked test for survival analyses to identify clinically associated deregulated circRNAs (Figure 1A). (2) The correlation between the expression of circRNAs and mRNAs is then determined using Pearson correlation to identify significant circRNA-mRNA pairs (|r| > 0.7, FDR < 0.05) (Figure 1A). (3) The clinically associated deregulated circRNAs are then integrated with significantly correlated circRNA-mRNA pairs to identify clinically relevant deregulated circRNA-mRNA pairs (Figure 1A).

The following steps were performed to generate a circRNA-miRNA-mRNA ceRNA network. (1) circRNA-miRNA and miRNA-mRNA pairs were first identified through integrating data from expression correlation analyses using Spearman correlation as well as prediction algorithms, namely miRanda [22] and PITA [23] to ensure that the circRNAs/mRNAs were not only predicted to be targets of the miRNAs, but their expressions were also inversely correlated with the corresponding miRNA. (2) The significance of sharing of miRNAs by the different ceRNAs (i.e., circRNA- circRNA, circRNA-mRNA and mRNA-mRNA) was then evaluated using various statistical analyses including the Hypergeometric test [24], Pearson correlation (PC) [24,25], partial Pearson correlation (PPC) [26], sensitive partial Pearson correlation (SPPC) [26], and conditional mutual information (CMI) [27]. (3) The relative confident ceRNA pairs that were found to be significant by PC (r > 0 & *p* < 0.05) and PPC methods (*p* < 0.05) were then integrated with clinically relevant deregulated circRNA-mRNA pairs (described earlier) to generate clinically relevant deregulated circRNA-miRNA-mRNA triplets. All of the analysis above was performed by using R and/or Matlab2017a.

The clinically relevant circRNA-mRNA co-expression and circRNA-miRNA-mRNA ceRNA networks were then generated using the Cytoscape software [28]. Pertinent cancer pathways that these circRNA-mRNA/circRNA-miRNA-mRNA ceRNA networks deregulate were then identified through their significantly correlated genes using ConsensusPathDB [29,30].

All the primers for RNAs, siRNAs for circRNAs and miRNAs, and probes for circRNAs are listed in Appendix A. Further details can be found in the Appendix A.

## 3. Results

### 3.1. Profiles of Differentially Expressed circRNAs, miRNAs and mRNAs in HCC Patients

Genome-wide circRNA, miRNA and mRNA expression profiles of tumorous tissue and non-tumorous tissue in 49 HCC patients were interrogated. 367/10,592 (~3.5%) circRNAs, 158/743 (~21.3%) miRNAs and 1983/17,261 (~11.5%) mRNAs were found to be significantly differentially expressed between tumorous tissue and adjacent non-tumorous tissue (Student’s t-test, FDR < 0.05, FC > 1.5|2). Unsupervised hierarchical clustering (Figure 1B) of 367 (236 upregulated, 131 downregulated) differentially expressed (DE) circRNAs, 158 (123 upregulated, 35 downregulated) DE miRNAs and 1983 (746 upregulated, 1237 downregulated) DE mRNAs revealed that tumorous tissue of HCC patients can be clearly distinguished from the non-tumorous tissue based on these profiles with only a few exceptions. The deregulated circRNAs were distributed throughout all the chromosomes except the Y-chromosome (Figure 1C), with the greatest proportion in chromosome 22. Most of the differentially expressed circRNAs (327/367 or 89.1%) were found to be transcribed from protein-coding exons (Figure 1D), consistent with previous observation [31].

### 3.2. Deregulated circRNAs, miRNAs and mRNAs Are Associated with Various Clinicopathological Features

To evaluate the clinical significance of these RNAs, clinical phenotypes were sub-grouped into three categories, namely, Tumor properties (Tumor size, Tumor grade, Encapsulation and Degree of encapsulation), Invasion and metastasis (Vascular invasion and Tumor invasion) as well as patients’ prognosis (Cancer stage and Overall survival) (Appendix A). A total of 132/367 (~36.0%), 96/158 (~60.8%) and 1008/1983 (~50.8%) deregulated circRNAs, miRNAs, and mRNAs, respectively, were found to be associated (|FC| > 1.5, *p* < 0.05, Student’s t-test) with clinical characteristics (Figure 1E). Most of the deregulated RNAs were associated with tumor properties, particularly tumor grade (14.7% circRNAs, 36.7% miRNAs and 24.0% mRNAs) (Figure 1D). In total, 29 circRNAs, 9 miRNAs and 196 mRNAs were associated with at least two clinical phenotypes (Figure 1F).

In total, 74 of the 132 circRNAs were potential oncogenic circRNAs since high tumor expression was consistently associated with poorer clinical characteristics (e.g., higher tumor grade), while 35 were potential tumor suppressor circRNAs since high expression of these circRNAs in tumors was consistently associated with better clinical characteristics (e.g., lower tumor grade) (Figure 2A).

### 3.3. Clinically Relevant Deregulated Co-Expressed circRNA-mRNA Networks

As circRNAs modulate mRNA expression, their functions may be inferred via “guilt-by-association” [32,33] through identifying mRNAs that are significantly co-expressed with circRNAs. Using Pearson correlation (|r| > 0.7, FDR < 0.05), 460 circRNA-mRNAs pairs were identified from 17 oncogenic circRNAs and 6 tumor suppressive circRNAs (Figure 2B, Appendix A). The 422 upregulated mRNAs that strongly correlated with the 17 oncogenic circRNAs were mainly enriched in cell cycle pathways (Figure 2C), while the 20 downregulated mRNAs strongly correlated with the 6 tumor suppressive circRNAs were mainly enriched in metabolism and metallothionein pathways (Figure 2D).

Cytoscape software [28] was employed to better visualize the relationship of the 460 clinically associated deregulated circRNA-mRNA co-expressed pairs. In total, 17 oncogenic (Figure 2E) and 6 tumor suppressive circRNAs (Figure 2F) clustered separately to form independent networks. The 17 oncogenic circRNAs were significantly co-expressed (|r| > 0.7, FDR < 0.05) with 159 genes forming two networks, namely a singleton (one circRNA-one mRNA) (Figure 2E(ii)) and a very large and complex network comprising 16 circRNAs and 158 genes (Figure 2E(i)). Notably, only five circRNAs (circGPC3/circR091581, circW7/circR001387, circW8/circR001388, circW3/circR103583, and circRaly/circR406097) in the large oncogenic circRNA-gene network are predicted to be sufficient to modulate the expression of 93% (147/158) of the genes in the network as these are significantly co-expressed with these 147 mRNAs (Figure 2E(i),G). In total, 89.8% of the co-expressed genes of these five nodal/master circRNAs were also associated with worse clinical characteristics, including high tumor grade or absent tumor capsule (Figure 2E(i),G). The six tumor suppressive circRNAs were significantly co-expressed with 20 mRNAs forming six distinct networks as shown in Figure 2F, four of which comprise only one or two genes. Of the two larger networks, co-expressed genes [11] of the largest network (Figure 2F(i)) were mainly metallothionein genes while the co-expressed genes [4] of the second largest network (Figure 2F(ii)) were found to be involved in butanoate, fatty acid and amino acid metabolism. These observations suggest that oncogenic and tumor suppressive circRNA-centred networks modulate prognosis of HCC patients through distinct pathways. Notably, the expression of these five nodal/master circRNAs with red and bold labels were also significantly correlated with each other (Appendix A), suggesting a potential built-in redundancy in pathway regulation. Interestingly, three of the oncogenic nodal circRNAs (circW7; circW8; circW3), together with four other circRNAs (circR103586, circR103582, circR103584, and circR103587) in the 16 circRNA-centred oncogenic networks and circR103585 were all derived from the same gene: WHSC1 (Figure 2E(i) and Appendix A). In fact, 48.5% of circRNAs were expressed in more than one isoform (Appendix A). On the other hand, expression of most of the six tumor suppressive circRNAs was less strongly correlated with each other (|r| < 0.6) (Appendix A) leading to less complex networks observed (Figure 2F).

To determine whether circRNA functions in cis through the parental mRNA/transcript or in trans through modulating other mRNAs, we evaluated whether the expression changes of the circRNAs were consistent with the host mRNAs. The expression changes of circRNAs and host genes between tumorous versus non-tumorous tissue were overall positively correlated (r = 0.277) (Appendix A, quadrant (ii) and (iii)), although ~2.5% of these circRNAs and their parental mRNA/transcripts have differential changes (Appendix A, quadrant (i) and (iv)), as evident from the examples given in the figure (Appendix A).

### 3.4. Oncogenic Roles of the Five Nodal/Master circRNAs Associated with Worse Clinical Characteristics

The oncogenic roles of the five nodal/master circRNAs (circGPC3, circW7, circW8, circW3, and circRaly) were experimentally evaluated since these key circRNAs are associated with clinical characteristics related to poorer patient prognosis and predicted to modulate the expression of ~93% (147/158) of the genes associated with the cell cycle pathway. Expression of all five nodal circRNAs were significantly associated (*p* < 0.05) with tumor grade with lowest expression in the non-tumorous tissue and highest expression in the higher grade tumors (Figure 3A). Expression of three of the five nodal circRNAs (circW7, circW8, and circW3) were also significantly associated (*p* < 0.05) with tumor capsulation, with non-tumorous tissue exhibiting the lowest expression, while the tumors with incomplete capsule, indicative of poorer prognosis, exhibited the highest expression (Figure 3A). Higher expression of circRaly was also significantly associated with poorer overall survival (Figure 3A). The higher expression of these nodal circRNAs in the tumors was validated using RT-qPCR of samples in independent cohort of HCC patients (Figure 3B).

To further characterize the roles of these nodal circRNAs in HCC, we evaluated their expression in a non-transformed liver cell-line, LO2, and a transformed cell-line, Huh7. All five circRNAs were expressed at higher levels in Huh7 compared to LO2 (Figure 3C). The junction of all five nodal circRNAs was identified through sequencing of the amplified products using the divergent primers (Appendix A). All five nodal circRNAs are resistant to Rnase R digestion (Appendix A), suggesting that the expression observed is circular RNA and not its linear host gene. All five nodal circRNAs were found to be primarily localized to cytoplasm as evident from sub-cellular fractionation (Appendix A). RNA immunoprecipitation (RIP) with Ago2-antibodies on Huh7 cells, which expresses high levels of all five nodal circRNAs, revealed that all five circRNAs can interact with Ago2 (Appendix A), suggesting that these may be silenced by miRNAs since Ago2 protein are essential components of the RNA-induced silencing complex (RISC) and play key roles in RNA silencing. This may also be evidence of their sponging ability since Ago2 is also involved in miRNA sponging.

### 3.5. The Five Nodal circRNAs Modulate Tumorigenesis by Promoting Proliferation and Anchorage Independent Growth

Introduction of circGPC3, circW3, circW7 and circRaly constructs into LO2 (the non-transformed cell-line) led to increased soft-agar colony formation, while attenuating the expression of circW8 in Huh7 led to significantly less colony formation (Figure 3D,E), confirming the oncogenic role of these five nodal circRNAs. Overexpressed circW3, circW7 and circRaly were also found to promote soft-agar colony formation in Huh7 (Appendix A).

Overexpressed nodal circGPC3, circW3 and circW7 in LO2 were found to significantly increase the cell proliferation, leading to shortened doubling time (Figure 3D,F), while knockdown of circW8 in Huh7 significantly inhibited cell proliferation, increasing their doubling time (Figure 3D,F). A similar trend was observed for circRaly, although it did not reach statistical significance (Figure 3D,F and Appendix A).

RNA sequencing was performed for the five key nodal circRNAs that were overexpressed or inhibited in non-transformed LO2 and/or transformed Huh7. In total, 98 upregulated and 96 down regulated genes were co-regulated by all five key nodal circRNAs (Appendix A). These were predicted to mainly co-upregulate signal transduction/immune system pathways (Appendix A), and co-downregulate organelle biogenesis/maintenance and metabolism pathways (Appendix A). Furthermore, the key nodal circRNAs were also found to upregulate genes involved in the deadenylation of mRNA (Appendix A), which is implicated in miRNA function [34], affirming their roles as sponges for miRNAs.

### 3.6. Clinically Relevant Differentially Expressed circRNA-miRNA-mRNA ceRNA Networks

circRNAs can modulate gene expression through various modalities, and one of the major ways they do so is by acting as competing endogenous RNAs (ceRNAs) or miRNA sponges/inhibitors to modulate the expression of mRNAs by competing for shared miRNAs. ceRNA networks were thus computationally inferred from circRNA, miRNA and mRNA expression profiles. Correlation (Spearman) analyses coupled with prediction (miRanda and PITA) algorithms (Figure 1A) were employed to identify 773 shared circRNA-miRNA pairs and 22,233 shared miRNA-mRNA pairs (Figure 4A, red circle). The significance of sharing the same miRNA response element (MRE) by circRNA/mRNA for each ceRNA pair was evaluated using the hypergeometric test [24]. A total of 784, 838,878 and 27,107 significant circRNA-circRNA, mRNA-mRNA and circRNA-mRNA pairs, respectively, were identified. In total, 0, 1679 and 41 highly confident circRNA-circRNA, mRNA-mRNA and circRNA-mRNA ceRNAs respectively were identified through integrating PC [24,25], PPC [26], SPPC [26] and CMI [27] statistical analyses (Figure 4B, red circle).

The 41 circRNA-mRNA ceRNA pairs generated 406 potential circRNA-miRNA-mRNA triplets (Appendix A). In total, 12 of the genes in these triplets which were upregulated in the tumors of HCC patients were mainly enriched in the cell-cycle, signaling pathways and carbohydrate metabolism, while 26 genes in these triplets were downregulated and enriched in genes associated with platelet activation, signaling, aggregation and degranulation (Figure 4C).

None of the 5 nodal circRNAs in the key oncogenic clinically relevant circRNA-mRNA network (Figure 2E), or any of the other circRNAs in the same network, were amongst the 41 highly confident circRNA-mRNA ceRNA pairs. We thus further evaluated if any of the five nodal circRNAs are predicted to compete with their co-expressed mRNA for miRNA binding at a lower stringency, by identifying moderately confident ceRNAs. The moderately confident ceRNAs were identified through the integration of only the PC and PPC statistical analyses, and 114,419 moderately confident circRNA-miRNA-mRNA triplets were derived. The 114,419 moderately confident circRNA-miRNA-mRNA triplets were intersected with the 460 clinically relevant circRNA-mRNA pairs (|r| > 0.7) (Figure 2B and Appendix A) to yield 34 circRNA-miRNA-mRNA triplets in the clinically relevant circRNA-miRNA-mRNA ceRNA network (Figure 4D,E and Appendix A). Notably, one of the nodal circRNAs in the oncogenic clinically relevant circRNA-mRNA network, circGPC3 (circR091581) (Figure 4Dii, red circle), was found to compete with 11 mRNA/genes (ASPM, CENPW, KIF14, NEK2, POLQ, TOP2A, DBF4, ERCC6L, E2F7, GPC3, and MMS22L) involved in cell-cycle for two miRNAs (miR-378c and miR-378a-3p) (Figure 4F).

### 3.7. Role of circGPC3 in Modulating Tumorigenesis through Sponging miR-378a-3p to Upregulate ASPM

Of the five nodal circRNAs, circGPC3 was predicted in silico to reside in a clinically-relevant circRNA-miRNA-gene ceRNA network and compete with several mRNAs for binding to two miRNAs (Figure 4Dii). Its host gene, GPC3 (Glypican 3), was implicated in tumorigenesis through the Wnt signaling pathway [35]. Hence, we further characterized cirGPC3 to understand how circGPC3 modulates cancer phenotypes leading to higher tumor grade and poorer cancer prognosis.

circGPC3 is derived from exon 3 of GPC3 (chrX:132887508-132888203). Expression of circGPC3 and its host gene, GPC3, were upregulated in the tumor tissue compared to the adjacent non-tumorous liver tissue in both the 47 discovery phase HCC samples and 15 independent cohort HCC samples (Appendix A and Figure 3B).

Overexpression of circGPC3 in LO2 cells and inhibition of circGPC3 in Huh7 cells upregulated the expression of 467 genes in the Toll-like receptors (TLRs) and transcription pathways (Appendix A), while downregulating the expression of 409 genes in the transcription and DNA repair pathways (Appendix A).

Interestingly, modulating the expression of circGPC3 altered the protein (Appendix A) but not the transcript (Appendix A) expression of its host gene GPC3. Inhibiting circGPC3 expression not only inhibited GPC3 protein expression, but it also inhibited β-catenin expression (Appendix A), which is consistent with previous observation of β-catenin acting downstream of GPC3 [36]. circGPC3 is predicted in silico to have potential to be translated (Appendix A) as a peptide, raising the possibility that circGPC3 may modulate phenotypes as a protein.

circGPC3 can be amplified using divergent primers from cDNA but not genomic DNA, suggesting that it is in the RNA form (Figure 5A). Fluorescence in-situ hybridization (FISH) revealed that circGPC3 mainly resides in the cytoplasm (Figure 5B). Live cell imaging showed that inhibition of circGPC3 expression in Huh7 significantly inhibited cell proliferation increasing their doubling time, while overexpression of circGPC3 in LO2 significantly increased the cell proliferation (Figure 3F and Figure 5C,D). These observations are consistent with the predicted oncogenic role of circGPC3. Cell-cycle analyses revealed that introduction of circGPC3 in LO2 led to more cells entering S/G2 phases and fewer in G0/G1 phases (although not significant), while the reverse is observed in Huh7 cells whose circGPC3 expression is inhibited with siRNAs (Figure 5E). Notably, attenuating the expression of circGPC3 in Huh7 cells led to significantly less soft-agar colony formation (Figure 5F).

Inhibition of circGPC3 expression in Huh7 led to reduced invasive ability (Figure 5G) and slower cell migration (Figure 5H), while over-expression of circGPC3 in non-transformed LO2 cells led to enhanced invasive capability (Figure 5G) and more rapid cell movement (Figure 5H). These observations in Huh7/LO2 cells were also validated in HepG2 and SNU449 cells (Appendix A), except that the attenuation of circGPC3 expression in HepG2 cells led to faster wound healing (Appendix A). This may be due to inhibition of circGPC3 expression upregulating genes in the VEGFA pathway in HepG2 but downregulating different genes in the same VEGFA pathway in Huh7 cells (Appendix A). Nonetheless, as circGPC3 also plays a role in cell proliferation, the observation may also be due to the effect of cell proliferation. Further experiments with mitomycin C to eliminate the effect of cell proliferation on wound healing will clarify this.

As circGPC3 was predicted in silico to reside in a clinically relevant circRNA-miRNA-gene ceRNA network and compete with 11 mRNAs for binding to miR-378a-3p (Figure 4Dii), we thus evaluated if the expression of these genes is affected by circGPC3. Using quantitative real-time RT-PCR (qRT-PCR), the trend of a few mRNAs is consistent with the observation in patients (Figure 6A) where their expressions are upregulated in LO2 cells transfected with circGPC3, and downregulated in Huh7 cells transfected with siRNA against circGPC3. As the ASPM gene showed the highest upregulation in LO2 and deepest downregulation in Huh7 cells, we further characterized the relationship between ASPM and circGPC3 experimentally. We hypothesized that circGPC3 competes with ASPM for binding to miR-378a-3p, leading to the modulation of cellular cancer phenotypes (Figure 6A).

We initially examined if miRNA-378a-3p is a target of both circGPC3 and ASPM by genetically engineering luciferase reporter constructs carrying the following: wild-type circGPC3 (circGPC3 WT); mutant circGPC3 (circGPC3 Mut) with the single predicted miRNA-378a-3p binding site altered (Figure 6B, top panel); wild-type ASPM (ASPM WT); and mutant ASPM (ASPM Mut) with all three predicted miRNA-378a-3p binding sites in the protein coding sequence (CDS) altered in the same construct (Figure 6B, top panel).

These constructs were then co-transfected with either control miRNA or miRNA-378a-3p into HEK293K cells and luciferase activity was determined. As evident in Figure 6B (bottom panel), miRNA-378a-3p inhibited luciferase activity of cells carrying the WT forms of circGPC3 and ASPM but not the mutant forms of both circGPC3 and ASPM where the binding to miR-378a-3p was abrogated. Hence, miRNA-378a-3p targets both circGPC3 and ASPM to inhibit their expression.

As Ago2 is an essential component of the RNA-induced silencing complex (RISC) and plays key roles in RNA silencing, we then determined if both circGPC3 and miR-378a-3p can bind to Ago2 as part of this complex. RNA immunoprecipitation (RIP) was performed on Huh7 cells, which expresses high levels of circGPC3. Endogenous circGPC3 and miR-378a-3p were immunoprecipitated from Huh7 cells with Ago2-antibodies (Sigma Aldrich, Saint Louis, MO, USA) and their expression was analysed using qRT-PCR. Figure 6C shows that both circGPC3 and miR-378a-3p were precipitated with the Ago2-antibodies, suggesting that they are in the RISC complex. We then proceeded to determine whether circGPC3 directly binds miR-378a-3p. As shown in Figure 6D, higher enrichment of circGPC3 and miR-378a-3p transcripts was observed, when Huh7 and circGPC3-expressing LO2 cell-lysates were probed with 3′-terminal-biotinylated-circGPC3 compared to control probes (Huh7 cells) or vector control (circGPC3-expressing LO2 cells), suggesting that circGPC3 interacted directly with miR-378a-3p. We then evaluated whether circGPC3 modulates the hallmarks of cancer in LO2 cells, through sponging miR-378a-3p, leading to the upregulation of ASPM expression. As shown in Figure 6E (top panel), the transformation potential of circGPC3-expression LO2 was significantly attenuated in the presence miR-378a-3p (histogram bars 1 versus 3) but not the control miRNA (histogram bars 1 versus 4). Similar trends were observed with cell proliferation (Appendix A) and cell migration (Appendix A) experiments in LO2 cells. Cells expressing circGPC3 were found to express higher levels of ASPM at the transcript levels (Figure 6E, middle panel, histogram bars 1 versus 2) as well as the protein levels (Figure 6F, right panel) in LO2 cells. On the other hand, inhibiting circGPC3 in Huh7 cells with siRNA against circGPC3 inhibited ASPM protein expression (Figure 6F, left panel). These data are consistent with previous reports that ASPM is associated with vascular invasion, early recurrence, and poor prognosis in HCC [37] and suggests that circGPC3 may modulate cancer phenotypes through the miR-378a-3p-ASPM pathway.

### 3.8. Significance of Clinically Relevant circRNA–Centric Regulatory Network in HCC

In summary, we propose a model for a clinically relevant circRNA-centric regulatory network in HCC (Figure 7A) by integrating clinical characteristics with circRNA, miRNA and mRNA expression computationally and validating experimentally.

From the receiver operating characteristic (ROC) analysis, all five key nodes showed excellent performance for distinguishing tumor grade (3,4) groups from adjacent non-tumorous tissue (*p* < 0.001, AUC > 0.9) (Figure 7B). circW8 and circRaly showed good performance for distinguishing tumor grade (3,4) from tumor grade (1,2) groups (*p* < 0.001, AUC > 0.8), while circW3 and circW7 showed fair performance (*p* < 0.01, AUC > 0.7) (Figure 7B). Although circGPC3 showed poor performance for distinguishing tumor grade (3,4) groups from tumor grade (1,2) groups (*p* < 0.05, AUC = 0.69), it was able to reasonably distinguish tumor grade (1,2) groups from adjacent non-tumorous tissue (*p* < 0.001, AUC > 0.8), similarly to circW3 and circW7 (Figure 7B). Logistic regression analysis showed that combination of circGPC3, circW3, circW7 and circRaly can excellently distinguish tumor grade (1,2) groups from adjacent non-tumorous tissue (*p* < 0.001, AUC = 0.92) with a high sensitivity of 0.82 and a high specificity of 0.95 (Figure 7C, bottom left), which suggested the potential roles of the key nodes in the early detection of HCC. Additionally, the combination of circGPC3, circW7, circW8 and circRaly had good performance for distinguishing tumor grade (3,4) groups from tumor grade (1,2) groups (*p* < 0.001, AUC = 0.88) (Figure 7C, bottom right). Hence, these nodal circRNAs have potential to be good prognostic biomarkers. Notably, the key nodal circGPC3 circRNAs were found to be a promising non-invasive biomarker for detecting HCC or even early HCC, as its expression is highest in the plasma of patients with higher grade tumors (3,4), compared to those with lower grade tumors (1,2), and lowest in healthy individuals (Figure 7D).

## 4. Discussion

Current limitations in HCC diagnosis/therapy necessitates the identification of novel molecules through better elucidation of the molecular network mechanisms of HCC. Here, based on circRNA, mRNA and miRNA expression profiles of HCC patients, clinically relevant circRNA-centric networks were computationally predicted. As illustrated in Figure 7A, the prognostic significance of these predicted circRNA-centric networks was evident from key cancer pathways molecules in these networks are predicted to reside in; the experimental demonstration of tumorigenic/metastatic phenotypes of the master/key nodal circRNAs; as well as the relevant cancer-related mechanism, through which the nodal circGPC3 cirRNA acts as a sponge of miR-378a-3p to alter the expression of ASPM (a cancer molecule [37]), leading to the modulation of tumorigenesis.

Seventeen circRNAs residing in two circRNA-mRNA networks were associated with clinical characteristics that relate to worse prognosis in HCC patients (higher tumor grade and less tumor encapsulation). Six circRNAs, residing in six separate circRNA-mRNA networks, were associated with clinical characteristics of better prognosis (smaller tumor size, absent tumor capsule and better overall survival). Encapsulated HCC tumors are less aggressive, with lesser invasion events [38]. Tumor grade was reported to be associated with deregulation of proliferation, cell cycle, metastasis and invasion-associated genes in HCC [39,40,41,42].

Five key/master nodal circRNAs (circGPC3, circW3, circW7, circW8 and circRaly), accounting for ~93% (147/158) of the co-expressed genes within the large 16-member clinically relevant worse prognosis circRNA-mRNA network, were identified. Notably, all five master circRNAs are significantly correlated with one another (Appendix A), suggesting a potential built-in redundancy of circRNAs in pertinent pathway regulation. Interestingly, three of these (circW3, circW7 and circW8) are derived from the same parental gene: WHSC1 (Appendix A). In fact, another circRNA (circ-NSD2), also derived from WHSC1, was reported to promote migration, invasion and metastasis of colorectal cancer (CRC) cells in a mouse model of CRC liver metastasis [43], highlighting that this family of circRNAs derived from WHSC1 may be worth further investigation.

To facilitate the design of therapeutic strategies for HCC based on circRNAs in key cancer-related circRNA-mRNA networks, it is pertinent to elucidate the molecular mechanisms by which circRNAs in these cirRNA-mRNA networks modulate tumorigenesis. While circRNAs can modulate gene expression through diverse ways, one of the major ways is by acting as ceRNAs/miRNA sponges to influence gene expression through competition for shared miRNAs [44]. Here again, through computational analyses, we identified a confident clinically relevant circRNA-miRNA-mRNA network that hosts circGPC3, one of the five key nodal circRNAs.

circGPC3 is the most highly upregulated circRNA in the tumors of HCC patients, and its high expression is significantly associated with a high tumor grade. High circGPC3 expression promoted cell proliferation, cell cycle, cell transformation, migration and invasion in both transformed and non-transformed liver cell lines (Figure 3E and Figure 5C–H). circGPC3 was computationally predicted to modulate expression of 11 genes by competing with two miRNAs, eight of which were experimentally validated (Figure 6A). Notably, we were able to experimentally demonstrate the circGPC3/miR-378a-3p/ASPM axis through luciferase assays, Ago2, RIP and circRNA pull-downs, as well as cancer phenotype assays (Figure 6). Our data are consistent with previous reports that both miR-378a-3p [45] and ASPM [37,46,47] can modulate cellular proliferation, migration and invasion. In fact, ASPM was previously reported to be over-expressed in HCC, and was highlighted as a biomarker associated with poorer prognosis in HCC [37,48] and breast cancer [49]. This study thus highlights circGPC3 as the link between miR-378a-3p and ASPM in their role in tumorigenesis and modulating HCC prognosis.

The ability of circRNAs (e.g., circGPC3) to sponge miRNAs (e.g., miR-378a-3p), to modulate tumorigenesis, suggests their potential as prognostic biomarkers and therapeutic targets [50]. In fact, the nodal circRNAs, circGPC3, circW3 and circW7 displayed good performance for distinguishing low tumor grades from adjacent non-tumorous tissue (*p* < 0.001, AUC > 0.8) (Figure 7B), suggesting the potential roles of these key nodal circRNAs in the early detection of HCC. circRNAs represent attractive minimally invasive biomarkers as they are present in various bodily fluids (including blood and saliva [51]) and are insensitive to ribonuclease, hence they are more stable than other linear RNA molecules due to the special circular closed structure [52]. Notably, plasma levels of circGPC3 were lowest in healthy individuals and highest in HCC patients with higher grade tumors (Figure 7D), suggesting their potential as promising minimally invasive prognostic biomarkers for HCC, although validation in a larger cohort of samples is necessary.

circRNAs also represent attractive targets for therapy, as targeting the unique back-splice junction of oncogenic circRNAs led to higher specificity with less off-target effects [44]. Interestingly, circGPC3 upregulated the protein but not the mRNA expression of its host gene GPC3 (Appendix A). circGPC3 was predicted by circBank [53] to have potential to be translated into protein (Appendix A), suggesting the circGPC3 may be bi-functional. On the one hand, circGPC3 can act at the RNA level to modulate the expression of its co-expressed genes (e.g., ASPM), perhaps as a miRNA sponge (Figure 6). On the other hand, we hypothesize circGPC3 may be translated into protein and influence the protein expression (Appendix A) of its host GPC3 gene, perhaps through affecting protein stability. In fact, high GPC3 protein expression was reported to be associated with poor prognosis of HCC patients [54]. circGPC3 was also found to alter the protein expression of β-catenin (Appendix A), which is consistent with reports that GPC3 can activate the targeted therapeutic Wnt signaling [55]. GPC3 is an established oncofetal glycoprotein that is highly overexpressed in HCC. It is hailed not only as a diagnostic [56] and prognostic [57,58] biomarker, but also as a novel, attractive therapeutic target that is currently in clinical trials [56,59]. It is thus tempting to propose circGPC3, which acts upstream GPC3, as another attractive prognostic biomarker which can also be developed as another novel therapeutic target. Further studies are necessary to validate if circGPC3 can act as a protein to modulate its host GPC3 protein expression as well as the Wnt pathway.

## 5. Conclusions

In summary, these data suggest the robustness of bioinformatically-predicted clinically relevant circRNA-mRNA networks and key/master nodal circRNAs, and underscore the important roles that these identified deregulated key/master circRNAs play in HCC. It also highlights the robustness of our computational strategy to identify clinically relevant circRNA-miRNA-mRNA ceRNA networks, which can be experimentally validated, e.g., circGPC3/miR-378a-3p/ASPM regulated axis. Indeed, circGPC3 has great potential to be a promising non-invasive prognostic biomarker for early HCC.

## Figures and Tables

**Figure 1 cancers-13-02772-f001:**
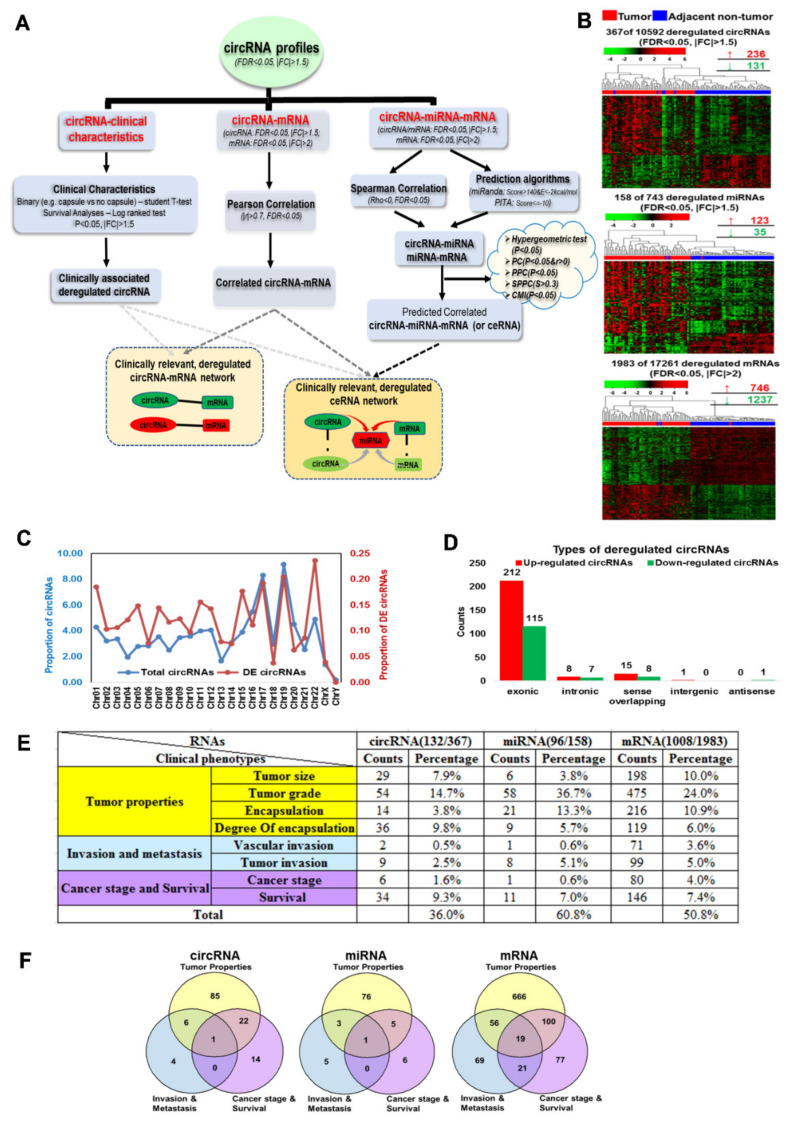
Three types of deregulated RNAs (circRNAs, miRNAs and mRNAs) and their clinical associations in HCC: (**A**) Schematic overview of the workflow implemented for identification of clinically relevant deregulated co-expressed circRNA-mRNA as well as circRNA-miRNA-mRNA competing endogenous RNA (ceRNA) networks. (**B**) Unsupervised hierarchical clustering analysis by Euclidean distance of deregulated circRNAs, miRNAs and mRNAs between 49 paired tumorous tissue and adjacent non-tumorous HCC tissue. (**C**) The proportion of total circRNAs (blue line) and differentially expressed circRNAs (red line) across human chromosomes. (**D**) Types of deregulated circRNAs. (**E**) Overall statistics for clinical features significantly associated with circRNAs, mRNAs and miRNAs. (**F**) Venn diagram showing the relationship of different clinical characteristics associated with deregulated circRNAs, miRNAs and mRNAs. For color in (**E**,**F**): Yellow: Tumor properties (including Tumor size, Tumor grade, Encapsulation and Degree of encapsulation); Blue: Invasion and metastasis (including Vascular invasion and Tumor invasion); Purple: Cancer stage and Survival.

**Figure 2 cancers-13-02772-f002:**
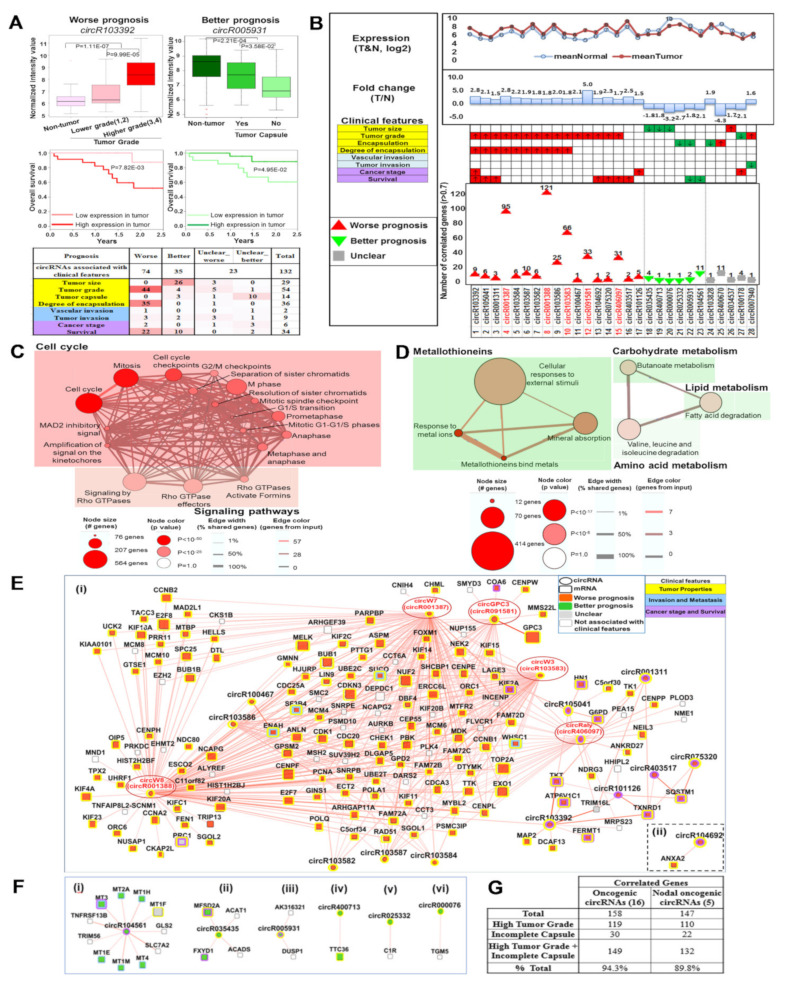
Prognostic circRNA-centric network modulating cell cycle and metabolism: (**A**) Prognostic RNA (e.g., circRNA) is defined as RNA that is differentially expressed in tumorous tissue vs. adjacent non-tumor, and is associated with worse/better clinical characteristics. Worse prognostic RNA is RNA whose expression is higher in tumors and the higher tumor expression is associated with clinical outcomes (e.g., higher grade tumors, absence of tumor capsule, worse overall survival) while better prognostic RNA is one whose expression is lower in tumors and the low tumor expression is associated with worse clinical outcomes. Examples of worse (left) and better prognostic cirRNAs (right) associated with tumor grade (top left), tumor capsule (top right) or overall survival (bottom) are shown. (**B**) Distribution of prognostic circRNAs that are significantly correlated with mRNAs (|r| > 0.7), including 17 worse prognostic circRNAs, 6 better prognostic circRNAs and 5 unclassified circRNAs. (**C**) Genes that are correlated with the 17 worse prognostic circRNAs reside mainly in the cell cycle and Rho GTPase signaling pathways. (**D**) Genes that are correlated with the six better prognostic circRNAs are mainly metallothionein genes or are involved in metabolism. (**E**) Clinically relevant deregulated co-expressed circRNA-mRNA network for the 17 worse prognostic circRNAs. The 5 nodal circRNAs are circled in red. (i) A large and complex network comprising 16 circRNAs and 158 genes. (ii) A singleton (one circRNA-one mRNA) network. (**F**) Clinically relevant, deregulated co-expressed circRNA-mRNA network for the six better prognostic circRNAs. (i) The largest network comprising one circRNA and 11 genes. (ii) The second largest network comprising one circRNA and 4 genes. (iii) The third largest network comprising one circRNA and 2 genes. (iv)(v)(vi) The singleton (one circRNA-one mRNA) networks. Ellipse: circRNAs; Rectangle: mRNAs; Red: Worse prognosis; Green: Better prognosis; Grey: Unclear prognosis; White: Not associated with any clinical features; outlines of nodes highlight the associated clinical features. Yellow: Tumor properties (including Tumor size, Tumor grade, Encapsulation and Degree of encapsulation); Blue: Invasion and metastasis (including Vascular invasion and Tumor invasion); Purple: Cancer stage and Survival. Red edges: Positive correlation. (**G**) Number of genes correlated with either the 17 worse prognostic circRNAs or the 5 nodal worse prognostic circRNAs that are associated with the different clinical characteristics.

**Figure 3 cancers-13-02772-f003:**
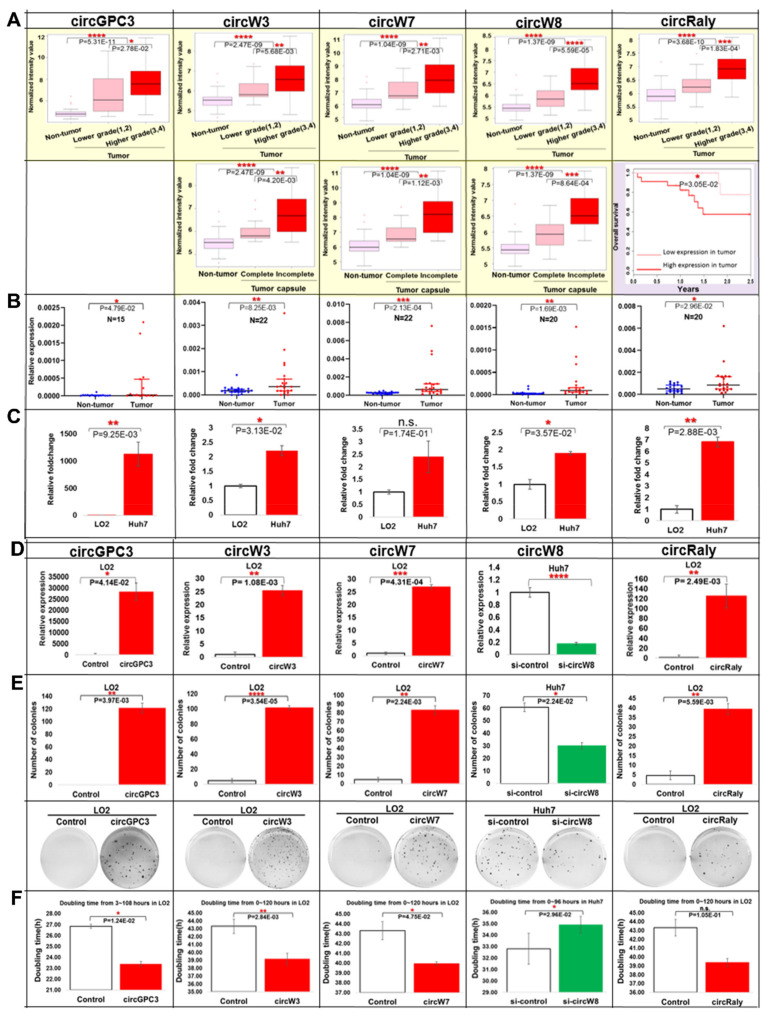
Five nodal circRNAs modulate tumorigenesis by promoting cell proliferation and anchorage independent growth: (**A**) Association of the five nodal circRNAs with clinical characteristics (tumor grade, tumor capsule and overall survival). (**B**) Relative expression of the 5 nodal circRNAs in independent cohort of HCC patients. Each colored dot represents the relative expression of either the tumorous (red) or non-tumorous (blue) liver tissue of each HCC patient. (**C**) Expression level of the 5 nodal circRNAs in LO2 and Huh7 cell lines. (**D**) Expression of the 5 nodal circRNAs in cell-lines transfected with the respective circRNAs (circGPC3, circW3, circW8, and circRaly) or si-RNA against circW8. (**E**) Top panel: Number of colonies in soft agar in cells transfected with the 4 nodal circRNAs or siRNA against circW8. Bottom panel: Representative figures taken from soft agar plates. (**F**) Doubling time of cells transfected with the 4 nodal circRNAs or siRNA against circW8. Data are from three independent experiments. Mean ± SEM (n.s.: not significant; * *p* < 0.05; ** *p* < 0.01; *** *p* < 0.001; **** *p* < 0.0001 by two-tailed Student’s *t*-test).

**Figure 4 cancers-13-02772-f004:**
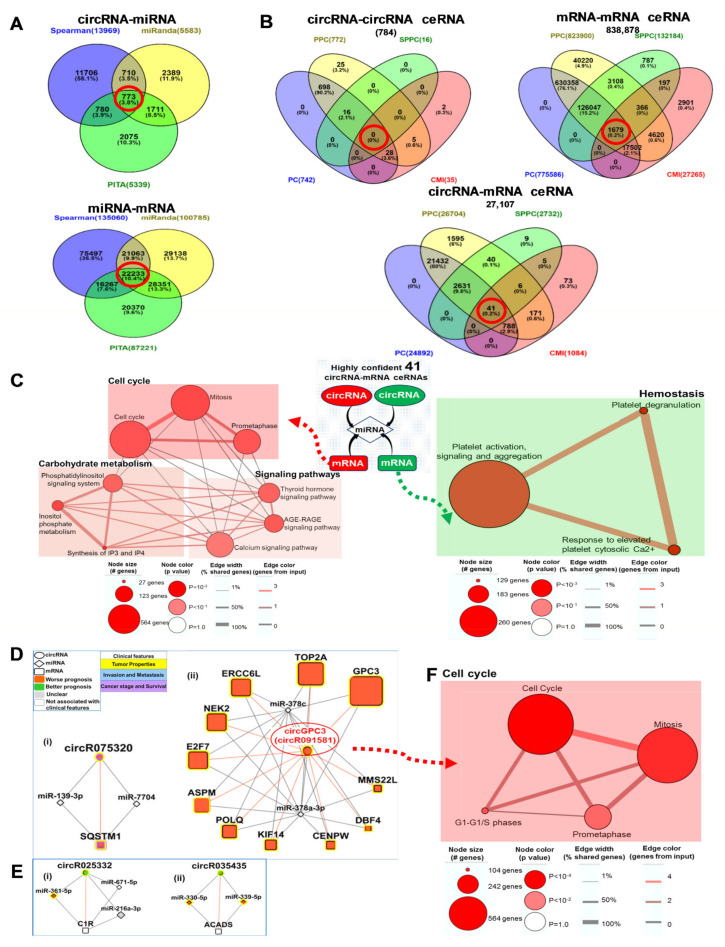
Clinically relevant differentially expressed circRNA-miRNA-mRNA ceRNA networks: (**A**) Venn diagrams showing the number of shared circRNA-miRNA (top) or miRNA-mRNA (bottom) that were determined through Spearman correlation analysis to be significant and/or predicted by miRanda and PITA algorithms to bind to each other. (**B**) Venn diagrams showing the number of circRNA-circRNA ceRNA (top left), mRNA-mRNA ceRNA (top right) and circRNA-mRNA ceRNA (bottom) that were found to show statistically confident interaction with either one or more than one of the four statistical analyses (Positive Correlation (PC), Partial Pearson Correlation (PPC), Sensitivity Partial Pearson Correlation (SPPC), Conditional Mutual Information (CMI)). Highly confident interaction is predicted by all statistical analyses/algorithms and is circled in red in each of the Venn diagrams. (**C**) Pathways of genes in the 41 highly confident circRNA-mRNA interactions. (**D**) Clinically relevant moderately confident (statistically significant based on PC and PPC statistical analyses only) circRNA-miRNA-mRNA ceRNA network associated with worse prognosis. Ellipse: circRNAs; Diamond: miRNA; Rectangle: mRNAs; Red: Worse prognosis; Green: Better prognosis; Grey: Unclear prognosis; White: Not associated with any clinical characteristics; outlines of nodes highlight the associated clinical features. Yellow: Tumor properties (including Tumor size, Tumor grade, Encapsulation and Degree of encapsulation); Purple: Cancer stage and Survival. Red edges: Positive correlation. Grey edges: Strength of correlation for circRNA-miRNA and miRNA-mRNA. (**E**) Clinically relevant moderately confident circRNA-miRNA-mRNA ceRNA network associated with better prognosis. Pathway of genes in the larger network (**Dii**) associated with circGPC3 is shown in (**F**).

**Figure 5 cancers-13-02772-f005:**
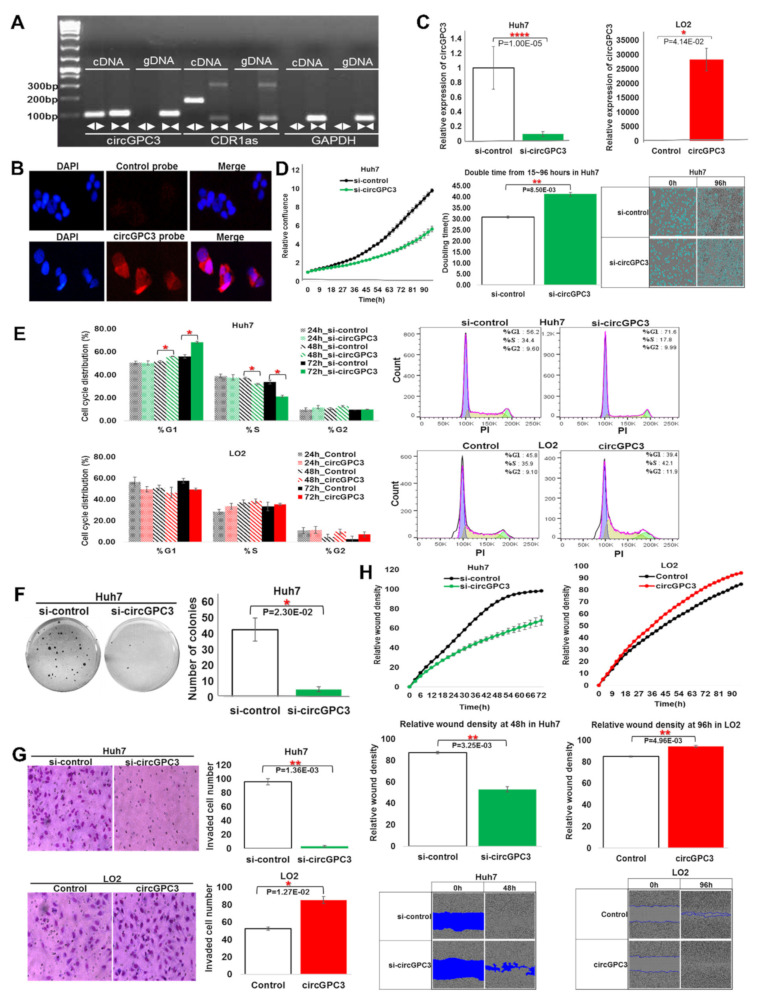
Experimental characterization of circGPC3: (**A**) Expression of circGPC3 determined through PCR using divergent and convergent primers in cDNA and genomic DNA. CDR1 was used as positive control with GAPDH as negative control. (**B**) Representative FISH (Fluorescent in situ hybridization) images demonstrating circGPC3 expression as detected using a junction probe that was labelled with biotin. (**C**) Expression of circGPC3 in Huh7 cells transfected with siRNA against circGPC3 (si-circGPC3) or in LO2 cells transfected with circGPC3. (**D**) Cell proliferation of Huh7 cells transfected with si-circGPC3. (**E**) Cell-cycle phase distribution of Huh7 and LO2 after knockdown (KD) and overexpression (OE) of circGPC3 respectively. (**F**) Soft-agar colony formation of Huh7 cells transfected with si-circGPC3. (**G**) Cell invasion assay of Huh7 and LO2 cells after KD and OE of circGPC3. (**H**) Cell migration for Huh7 and LO2 cells after KD and OE of circGPC3. Data are from three independent experiments. Mean ± SEM (n.s.: not significant; * *p* < 0.05; ** *p* < 0.01; **** *p* < 0.0001 by two-tailed Student’s *t*-test).

**Figure 6 cancers-13-02772-f006:**
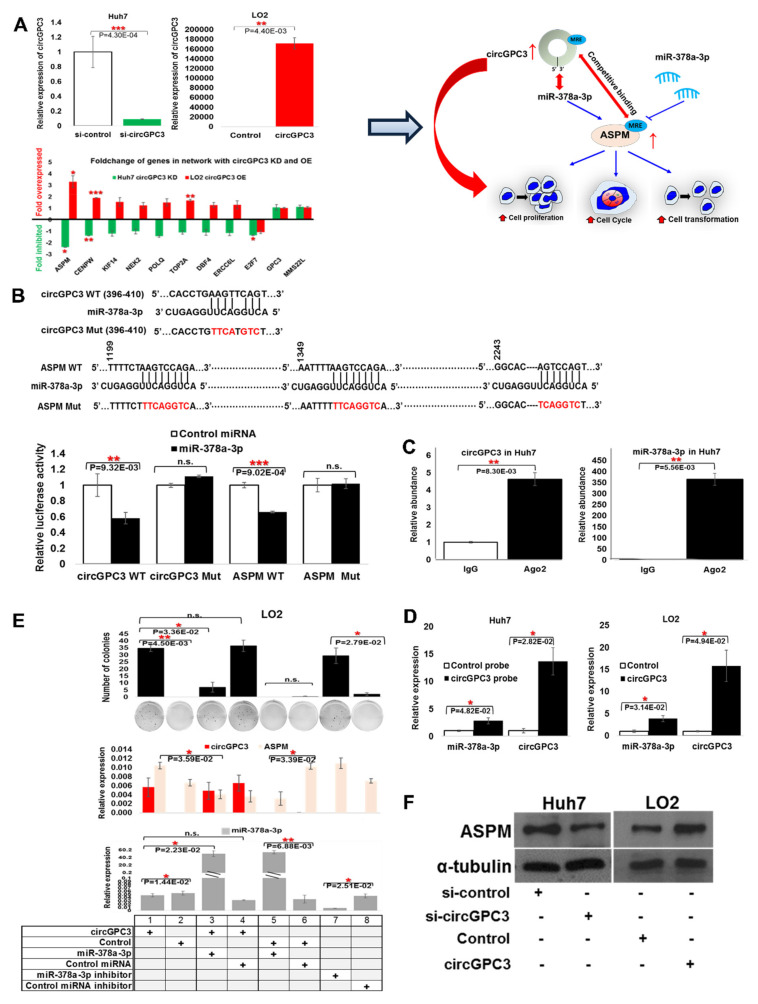
CircGPC3-centred ceRNA regulation through circGPC3/miR-378a-3p/ASPM axis: (**A**) Right: Validation of genes predicted to correlate with circGPC3 in clinically relevant circRNA-miRNA-mRNA ceRNA network (see Figure 5D). Left: Hypothesis of the role of circGPC3 in modulating various oncogenic phenotypes (e.g., cell proliferation, cell cycle and cell transformation) through the circGPC3/miR-378a-3p/ASPM axis. (**B**) Top: Sequence of Mutation (in red) generated for circGPC3 Mut and ASPM Mut (at the miRNA binding site within the protein coding sequence). Bottom: Luciferase assays with reporter constructs containing the wild-type or mutant circGPC3/ASPM downstream of a luciferase gene were performed after co-transfection with miR-378a-3p in HEK293T cells. (**C**) RNA Immunoprecipitation (RIP) experiments were performed using an antibody against Ago2 on extracts from Huh7 cells. Expression of circGPC3 and miR-378a-3p was detected. (**D**) Huh7 and circGPC3-expressing LO2 cell-lysates were probed with 3′-terminal-biotinylated-circGPC3 compared to control probes (Huh7 cells) or vector control (circGPC3-expressing LO2 cells). (**E**) circGPC3 promoted the cell transformation via the miR-378a-3p-ASPM pathway. (**F**) Protein level of ASPM after circGPC3 KD or OE. Data are from three independent experiments. Mean ± SEM (n.s.: not significant; * *p* < 0.05; ** *p* < 0.01; *** *p* < 0.001; by two-tailed Student’s *t*-test).

**Figure 7 cancers-13-02772-f007:**
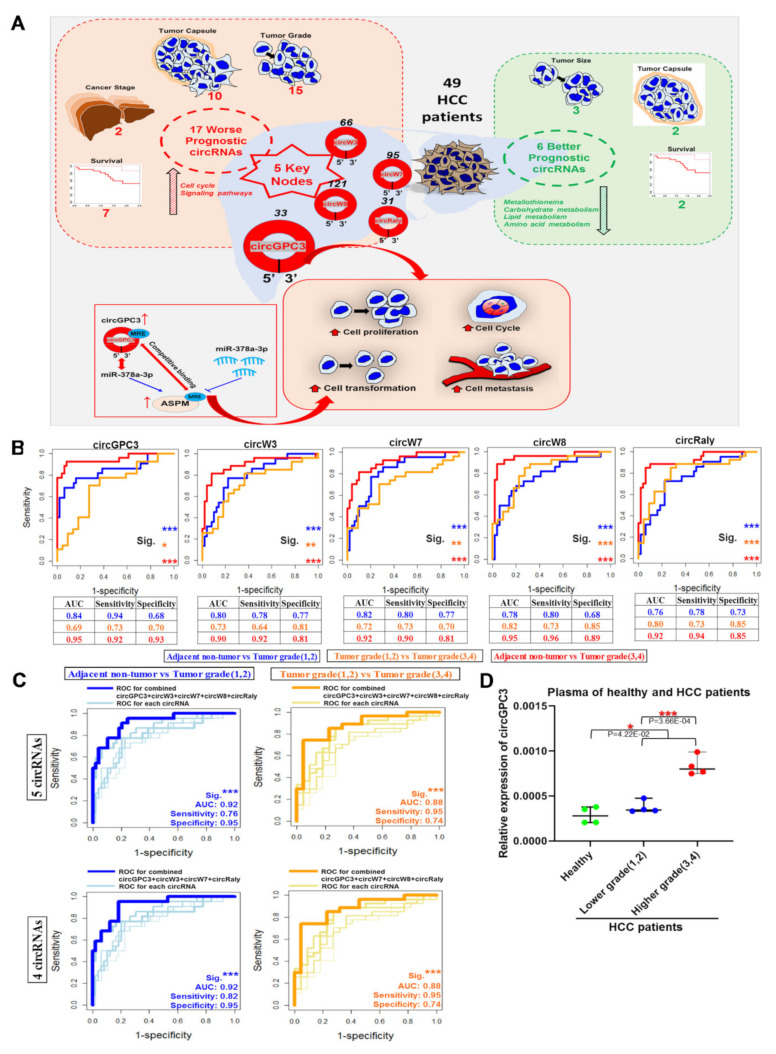
The clinical significance of a circRNA-centric regulatory network in HCC. (**A**) Model of clinically relevant circRNAs and networks associated with pertinent clinical characteristics and their role in modulating tumorigenesis. (**B**) ROC (Receiver operating characteristic) curve illustrating the diagnostic ability of the five nodal circRNAs to distinguish adjacent non-tumorous tissue from low tumor grade (1,2) (blue line), low tumor grade (1,2) from high tumor grade (3,4) (orange line) and adjacent non-tumorous tissue from high tumor grade (3,4) (red line). (**C**) Logistic regression and ROC curve analysis to evaluate the diagnostic ability of combined nodal circRNAs to distinguish adjacent non-tumorous tissue from low tumor grade (1,2) (blue line, left panel) and low tumor grade (1,2) from high tumor grade (3,4) (orange line, right panel). Top panel shows combination of all five nodal circRNAs while bottom panel shows combination of four of the five nodal circRNAs (except circW8). (**D**) Expression of circGPC3s in the plasma of healthy volunteers (green) or HCC patients (Blue: Low tumor grade (1,2); Red: High tumor grade (3,4)) (*n* = 4). Mean ± SEM (n.s.: not significant; * *p* < 0.05; ** *p* < 0.01; *** *p* < 0.001; by two-tailed Student’s *t*-test).

## Data Availability

The expression profiles for circRNAs, miRNAs and mRNAs in this study are deposited in Gene Expression Omnibus with series entry GSE155949, GSE156087 and GSE138178, respectively.

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
