# Peer review of "Clinicopathological-Associated Regulatory Network of Deregulated circRNAs in Hepatocellular Carcinoma"

_cancers, 2021, doi:10.3390/cancers13112772_

Round 1

Reviewer 1 Report

Han and Colleagues identified and discussed the potential role of circRNAs in HCC progression using a genome-wide approach that allowed the identification of a specific network. It was experimentally validated. The article is reach of information that can be useful for the scientific community but not all information is available. Moreover, figures can be improved to better understand results. In some cases, the statistic is lacking especially in supplementary figures. Below my comments.

Major

  1. Please consider to include in the introduction also other similar works as “Circular RNA Signature in Hepatocellular Carcinoma”. Is there an overlapping of circRNAs identified in the study previously cited with those that were identified in this new presented analysis? Moreover, several databases are published also describing HCC samples (e.g. CCRDB: a cancer circRNAs-related database and its application in hepatocellular carcinoma-related circRNAs) or network-based analyses (e.g. A circRNA–miRNA–mRNA network identification for exploring underlying pathogenesis and therapy strategy of hepatocellular carcinoma or Construction of circRNA-Based ceRNA Network to Reveal the Role of circRNAs in the Progression and Prognosis of Hepatocellular Carcinoma). Please improve the introduction with these information.  
  2. Why did the Authors decide to use different techniques to identify miRNA and circRNA or mRNA expression? Since the Authors used microarray for mRNA and circRNA expressions why not using the same technique also for miRNAs?
  3. In the Supplementary Materials, the Authors say that RNAse A treatment was done starting from 1.5 mg of total RNA. Is it correct or it is a mistake and they did it with 1.5 ug of total RNA?
  4. For all cloning experiments, please indicate restriction enzymes used, time, amount of DNA used for restriction reactions, and time and amount of DNA used in the ligation reaction. Moreover, in the table of the primers indicate accessory sequences to allow cloning and specific part for gene amplification via PCR. 
  5. In the Supplementary information Authors say “…were added to the upper chamber with 500 μl of DMEM with fetal bovine serum, and 1000 μl of DMEM medium with 10% FBS was added to the lower chamber.” Is the upper and lower medium the same or the concentration of FBS is different? Please specify.
  6. In Supplementary information, Authors say “The lysed cells were sonicated at ‘high’ setting ….” Which is the instrument used to understand what does it mean “high settings”?
  7. Please, include in the GEO platform all Raw and normalized data obtained from the microarray and sequencing experiments from HCC patients. This is useful to control the study and to allow researchers to develop the research to counteract hepatocellular carcinoma.
  8. Please, specify in the Methods in Supplementary information the software used to perform hierarchical clusters. Moreover, include in the caption of Figure 1 B which distance was used.
  9. According to me the association of Figure 1D in this phrase is a mistake: “ Most of the deregulated RNAs were 150 associated with tumor properties, particularly tumor grade (14.7% circRNAs, 36.7% miR- 151 NAs and 24.0% mRNAs) (Figure 1D).”
  10. Data in Figure 3 B, C and D represent the same concept (gene expression). Please indicate in the figure caption the difference in the visualization or use the same visualization type.
  11. Because it is impossible to appreciate SEM in Figure 3 C (panel 1, 5), D (panel 1, 2, 3, 5), E (panel 1, 2, 3) please consider rescaling the Y axes also introducing a breaking point. Moreover, change the Y axes for Figure 3C (panel 2, 3, 4) avoiding negative numbers that are not useful. 
  12. In Figure 3C and D the statistic is not indicated. Please indicate it.
  13. Some Supplementary Figures are lacking the indication of statistics (e.g. Figure S2B, S4 B, C…). Please include them. 
  14. In Tables S3, S5, and S6 indications of clinical features are not correctly aligned to the corresponding gene. Please consider loading an Excel file instead of a Word file. Moreover, for a simpler reading if the Authors do not load an Excel file repeat header table indications on each page when the table is divided into different pages. 
  15. Figure S5A represents an enrichment also for the deadenylation of mRNA that is important for miRNA functioning. Please consider to mention also this aspect that may be important for the action of circRNA as sponges.
  16. Why in Table S4 the same genes compare in the up and down expressed list (ZNF266, GLYR1, AMPD2, MFSD11, SLC25A45, C2orf42, DACT2, DMTN, SEC24B)? Table S7 has the same “problem”. 
  17. Please pay attention when compiling tables because the gene SEPT4 is changed in 4-Sep (e.g. Table S5). This may also affect GO analysis because the gene 2-Sep is not found in the databases.
  18. Please mitigate the sentence regarding the LO2 cells because differences are not significant (Figure 5E).
  19. Line 388: the cited Table S8 is not present in the supplementary data.
  20. Authors tested in qRT-PCR the expression of genes potentially regulated by miR-378a-3p. In line 392 they say that the expression of 8 on 11 tested gene targets were affected by the modulation of the expression of circGPC3. According to me, this sentence is misleading because observing Figure 5A probably genes ERCC6L, DBF4, NEK2, KIF14, and TOP2A are not statistically up- or down-regulated. Please consider performing a t-test between the control and the tested condition and discuss only genes statistically confirmed. Moreover, please consider moving the X labels below the green columns. Finally, explain the Y-axis. It is indicated as Fold Change, but it is impossible it is a fold change because a fold change is not negative if it is not logarithmic. Please, calculate the amplification efficiency for the primers used in the qRT-PCR experiments. What does MRE mean in Figure 5A?  

Minor.

  1.  In Supplementary information please change “…miRNA inhibitors for this studies as well as primers for ….” with “miRNA inhibitors for THESE studies as well as primers for ..”
  2. Please in the Supplementary information correct the sentence “….were measured continuously using a dual luciferase reporter assay system (Promega, WI, USA)according to the manufacturer’s instructions, The firefly luciferase activities were normalized to Renilla luciferase activity.” as following “were measured continuously using a dual luciferase reporter assay system (Promega, WI, USA){SPACE}according to the manufacturer’s instructions,{DOT} The firefly luciferase activities were normalized to Renilla luciferase activity. 
  3. Change 3F with 2F in the phrase “The six tumor suppressive circRNAs were significantly co-expressed with 20 mRNAs forming six distinct networks as shown in Fig. 3F”. Moreover, writhe Figure instead of Fig.
  4. Instead of “48.5% of circRNAs have >1 isoform” probably it is better “48.5% of circRNAs are expressed in more than one isoform.
  5. Authors suggest that since Ago2 interact with all the five nodal circRNAs they may be silenced by miRNAs. Ago2 is involved also in miRNA sponging therefore it should better to say that this may be evidence of their sponging ability (lines 261-262).

Reviewer 2 Report

In this study, Han et al., have performed a strategy to identify key circRNA signatures of circRNA-mRNA network for prognostic analysis of hepatocellular carcinoma (HCC) patients. By integration, circRNA-mRNA networks were established. By functional study, they found that most of nodal circRNAs are critical for HCC proliferation, transformation and cell invasiveness. Among these, circGPC3, which co-express with 33 genes and interacts/ compete with 11 mRNA with clinical significance.  circGPC3 was shown to act as a sponge of 29 miR-378a-3p to regulate APSM-mediated HCC transformation. To sump up, this study identified several circRNAs in a clinically-relevant, circRNA-centric network. This study is interesting and provide potential markers for prognosis and therapeutic intervention for HCC patients. Please find the following comments for further improvement of this manuscript:

  1. In Figure 3B, please explain why the sample number is 15 for circGPC3 analysis, while others are n=20 or n=22.
  2. For functional analysis, why circW8 was chosen for knockdown approach while its expression in Huh7 is higher than LO2. In addition, why Huh7 cells have been chosen for this analysis?
  3. In Figure 5C ,D and F, there is no Y axis in the figure.
  4. Since circGPC3 plays role in cell proliferation, the authors should add mitomycin C in order to eliminate the effect of cell proliferation for wound healing/ invasion assay.
  5. Figure 6C (X-axis) has not been shown properly.
  6. What is the effect of miR-378a-3p on expression of ASPM?

Round 2

Reviewer 1 Report

The authors revised the manuscript responding to almost all my comments. My last comments regard

  • the availability of genome-wide data. They are scheduled to be released on Aug 08, 2023. They should be released once the manuscript will be published.
  • qRT-PCR primer efficiency. It was not calculated. It is important to evaluate the consistency of qRT-PCR experiments. Please include it in the final manuscript.

Reviewer 2 Report

I have no further comments.

Author Response

Thanks a lot for the helpful comments and suggestions! 

Round 3

Reviewer 1 Report

The Manuscript can be published as it is.